# Electrostatic Adsorption of Dense AuNPs onto Silica Core as High-Performance SERS Tag for Sensitive Immunochromatographic Detection of *Streptococcus pneumoniae*

**DOI:** 10.3390/pathogens12020327

**Published:** 2023-02-15

**Authors:** Wanzhu Shen, Jiaxuan Li, Bo Jiang, You Nie, Yuanfeng Pang, Chongwen Wang, Rui Xiao, Rongzhang Hao

**Affiliations:** 1School of Public Health, Capital Medical University, Beijing 100069, China; 2Beijing Institute of Microbiology and Epidemiology, Beijing 100071, China; 3College of Life Sciences, Anhui Agricultural University, Hefei 230036, China

**Keywords:** immunochromatographic assay, SERS-ICA, SERS tag, *Streptococcus pneumoniae*, sensitive detection

## Abstract

*Streptococcus pneumoniae* (*S. pneumoniae*) is a prominent pathogen of bacterial pneumonia and its rapid and sensitive detection in complex biological samples remains a challenge. Here, we developed a simple but effective immunochromatographic assay (ICA) based on silica-Au core-satellite (SiO_2_@20Au) SERS tags to sensitively and quantitatively detect *S. pneumoniae.* The high-performance SiO_2_@20Au tags with superior stability and SERS activity were prepared by one-step electrostatic adsorption of dense 20 nm AuNPs onto 180 nm SiO_2_ core and introduced into the ICA method to ensure the high sensitivity and accuracy of the assay. The detection limit of the proposed SERS-ICA reached 46 cells/mL for *S. pneumoniae* and was 100-fold more sensitive than the traditional AuNPs-based colorimetric ICA method. Further, considering its good stability, specificity, reproducibility, and easy operation, the SiO_2_@20Au-SERS-ICA developed here has great potential to meet the demands of on-site and accurate detection of respiratory pathogens.

## 1. Introduction

*Streptococcus pneumoniae* (*S. pneumoniae*) is a dangerous respiratory pathogen that causes serious diseases such as pneumonia, sepsis, otitis media, and meningitis, and kills more than one million people worldwide a year [1,2,3]. *S. pneumoniae* easily affects children, the elderly, and immunocompromised people, and causes clinical symptoms (e.g., cough, fever, fatigued) resembling respiratory virus infection [4]. Timely and accurate diagnosis of *S. pneumoniae* is the key to guiding medication and saving lives. The current mature technologies for bacteria identification mainly include microbial cultivation, polymerase chain reaction (PCR), DNA sequencing, and mass spectrometry, which require laborious sample pretreatment, complex instruments, and long detection time (3–48 h) to output results, and thus suffer from being expensive and time-consuming [5,6,7,8]. Thus, a rapid and low-cost analytical technique for the sensitive detection of *S. pneumoniae* is desperately needed.

Given its outstanding advantages of simple operation, real-time analysis, user-friendliness, and low cost, the gold nanoparticle (AuNP)-based immunochromatographic assay (ICA) has dominated the point-of-care testing (POCT) market over the past few decades [9,10,11]. However, the inherent defects of colorimetric signals from AuNP including low sensitivity and limited quantitative ability have blocked the further application of the ICA method in trace substance detection [12,13,14]. To improve the sensitivity of ICA strip, in recent years, researchers have introduced several kinds of novel signal materials such as fluorescent microspheres, magnetic particles, nano-enzyme materials, and surface-enhanced Raman scattering (SERS) tags to replace the colorimetric signal nanomaterials [15,16,17,18,19,20]. Among these novel signals, SERS has been considered a powerful, fingerprint-specific vibrational spectroscopy, which can provide a non-destructive and ultrasensitive characterization of molecules on or near the surface of “free-electron-like” metal materials (mainly Au, Ag, and Cu) [21,22,23]. The newly developed SERS-based ICA strategy exhibits tremendous advantages including ultra-sensitivity (single molecular level), specific analysis (fingerprint spectrum), and highly stable signal (no photobleaching) for target quantification [24]. Technically speaking, the performance of SERS tags highly determined the sensitivity and stability of SERS-ICA methods [25,26]. Through many SERS nanostructures including colloidal metal NPs (e.g., AuNP, AgNP, and Au@Ag alloy) and Au/Ag coated nanocomposites (e.g., Fe_3_O_4_@Au, SiO_2_@Ag, and GO@Au) have been proposed for ICA analysis, these SERS tags usually face the problems of poor stability in complex samples (e.g., clinical samples, biospecimens, and foods), insufficient SERS activity, and uncontrollable hotspots, which weaken the analytical ability (accuracy, stability, and reproducibility) of SERS-ICA methods [27,28,29,30,31,32,33]. To date, a simple and efficient SERS-ICA method for respiratory bacteria detection has yet to be developed.

In this work, we reported a highly stable SERS-ICA approach by using novel silica-Au core-satellite (SiO_2_@20Au) nanocomposites as superior SERS tags for the rapid, sensitive, and quantitative determination of *S. pneumoniae*. The SiO_2_@20Au tag was simply fabricated by assembling one layer of dense 20 nm AuNPs onto the surface of 180 nm SiO_2_ via polyethyleneimine (PEI)-mediated electrostatic interaction, which can provide strong SERS activity, excellent colloidal and chemical stability, good dispersity, and multiple surface sites for bacteria binding. Under optimal conditions, the established SiO_2_@20Au-SERS-ICA can achieve direct detection of *S. pneumoniae* in 20 min with a detection limit of 46 cells/mL. Moreover, the proposed assay maintained its high accuracy, specificity, and stability in real biological samples. This behavior confirmed that the SiO_2_@20Au-SERS-ICA has great potential for the rapid and accurate monitoring of *S. pneumoniae* in respiratory tract samples.

## 2. Experimental Section

### 2.1. Chemicals, Materials, and Instruments

Tetraethyl orthosilicate (TEOS), branched PEI (MW ~25 kDa), chloroauric acid tetrahydrate (HAuCl_4_·4H_2_O), 5,5′-Dithiobis-(2-nitrobenzoic acid) (DTNB), N-(3-dimethyaminopropy)-N′-ethylcarbodiimide hydrochloride (EDC), 2-(N-morpholino) ethanesulfonic (MES), N-hydroxysulfosuccinimide sodium salt (sulfo-NHS), bovine serum albumin (BSA), and fetal bovine serum (FBS) were purchased from Sigma-Aldrich (St Louis, MO, USA). Goat anti-mouse IgG was supplied by Sangon Biotech Co., Ltd., (Shanghai, China). We purchased a pair of mouse monoclonal anti-*S. pneumoniae* antibodies from ACTHTEAM, LLC (Chicago, IL, USA). The ICA accessories (the sample pad, absorbent pad, and plastic backing card) were obtained from Jieyi Biotechnology Co., Ltd., (Shanghai, China). The CN95 nitrocellulose (NC) membrane was supplied by Sartorius (Gottingen, Germany).

The instruments for nanostructure characterization and SERS signal reading are described in Appendix A.

### 2.2. Preparation of 180 nm SiO_2_ NPs, 20 nm AuNPs, and SiO_2_@20Au NPs

The 180 nm SiO_2_ NPs with high dispersibility were fabricated by using a typical Stöber method [34]. Briefly, 3.5 mL of ammonia solution (~28%) was added to a mixture of ethanol/deionized water (100/6 mL), and the mixture was vigorously stirred at room temperature. Subsequently, 4 mL of TEOS was rapidly injected into the above mixture and the reaction was kept for 4 h. The formed SiO_2_ NPs were collected by centrifugation (6000 rpm, 6 min), rinsed twice with ethanol, and dried in an oven at 60 °C for future use.

The negatively charged AuNPs with an average diameter of 20 nm were synthesized through the typical trisodium citrate reduction method [35]. In brief, 1 mL of 1% HAuCl_4_ solution (*w*/*v*) was added to 100 mL deionized water, and the solution was heated to boiling. Then, 1.7 mL of trisodium citrate aqueous solution (1%) was quickly added into the boiling solution under vigorous stirring. After reaction for 15 min, the resulting 20 nm AuNPs were naturally cooled to room temperature and stored in a 4 °C refrigerator for future use.

The SiO_2_@20Au nanocomposites were fabricated by PEI-mediated electrostatic adsorption (Figure 1A). First, 1 mg of SiO_2_ NPs was dissolved in 10 mL deionized water, and then mixed with 10 mL of PEI aqueous solution (0.5%, *v*/*v*). Under sonication for 30 min, the PEI can be assembled rapidly onto the surface of SiO_2_ NPs to form SiO_2_@PEI. By centrifugal collection (6000 rpm, 6 min), the excess PEI in the supernatant was removed and the SiO_2_@PEI NPs were redispersed in 10 mL of deionized water. Second, the SiO_2_@PEI was directly reacted with 50 mL of 20 nm AuNPs (~10 nM) for 30 min under vigorous sonication. Notably, the AuNP suspension was not concentrated or diluted before use. The resulting SiO_2_@20Au NPs were separated by centrifugation (4500 rpm, 6 min) and stored in 10 mL ethanol for later use.

### 2.3. Preparation of SiO_2_@20Au SERS Tags

The SiO_2_@20Au SERS tags were fabricated by successively modifying DTNB molecules and anti-*S. pneumoniae* antibodies onto the SiO_2_@20Au surface, as illustrated in Figure 1B. In brief, 10 µL of DTNB (10 mM) ethanol solution was added into 1 mL of SiO_2_@20Au solution, and the mixture was vigorously sonicated for 2 h. Next, the DTNB-modified SiO_2_@20Au NPs (SiO_2_@20Au/DTNB) were separated by centrifugation (4500 rpm, 6 min), resuspended in 1 mL of MES solution (10 mM, pH 5.5), and then reacted with 5 µL of EDC (100 mM) and 10 µL of NHS (100 mM). After 15 min reaction, the activated SiO_2_@20Au/DTNB was collected by centrifugal force and incubated with 200 μL of PBS solution (10 mM, pH 7.4) and 10 μg of the detection antibody for *S. pneumoniae*. The mixture was shaken in an oscillator for 2 h at 800 rpm/min and then incubated with 100 μL of BSA (10 wt%) for another 30 min to block the excess activated sites of SiO_2_@20Au tags. Finally, antibody-conjugated SiO_2_@20Au SERS tags were separated by centrifugation (4500 rpm, 6 min), washed with 10 mM PBST buffer (pH 7.4, 0.05% Tween), and stored in 200 μL of preservation solution (10 mM PBST, 1% BSA (*w*/*v*) and 0.04% NaN_3_ (*w*/*v*)) for future use. 

### 2.4. Preparation of SiO_2_@20Au-Based SERS-ICA for S. pneumoniae Detection

The structure of the SERS-ICA strip for *S. pneumoniae* detection is shown in Figure 1C, which consisted of three independent components including a sample pad, an NC membrane with a test (T) line and a control (C) line, and an absorbent pad to provide the capillary driving force. The T line and C line were sprayed with 1 mg/mL of *S. pneumoniae* capture antibody and 1 mg/mL of goat anti-mouse IgG, respectively, with a dispense rate of 0.1 μL/mm via a spraying platform (Biodot syz5050). The antibody-loaded NC membrane was then dried at 37 °C for at least 3 h and assembled with the sample pad and absorbent pad onto a plastic backing card. The prepared ICA card was cut into individual 3.5 mm wide strips with an automatic programmable cutter and finally placed in a sealed bag with a desiccant for storage. 

### 2.5. Preparation of Bacterial Sample 

The standard strain of *S. pneumoniae* (ATCC 49619) was supplied by Prof. Bing Gu’s laboratory in Guangdong Provincial People’s Hospital and verified by PCR. The PCR primers for target gene lytA were displayed as follows [36]:
Forward primer: 5′-AACTCTTACGCAATCTAGCAGATGAA-3′; Reverse primer: 5′-CGTGCAATACTCGTGCGTTTTA-3′

The concentration of the *S. pneumoniae* sample was determined by the conventional plate counting method. In brief, *S. pneumoniae* was cultivated in 5% sheep blood agar plates at 37 °C in an atmosphere containing 5% CO_2_. After 12 h culture, dozens of colonies were picked from the plate and transferred into 1 mL of PBS solution (10 mM, pH 7.4) as the initial bacterial solution. Then, 0.1 mL of bacterial solution was diluted in sterile water 1 × 10^5^ and 1 × 10^4^ times and coated on the blood agar plate at 37 °C. After overnight culture, the number of colony-forming units (CFUs) on the plates was counted. According to the results of CFU counting, the initial bacterial solution was adjusted to concentrations of 10^7^ cells/mL for follow-up tests. The bacteria counting results are shown in the Appendix A.

### 2.6. Bacteria Detection via SiO_2_@20Au-Based SERS-ICA

To verify the performance of SERS-ICA based on SiO_2_@20Au tags, different concentrations (10^6^–10 cells/mL) of the *S. pneumoniae* were spiked into PBS solution and actual clinical samples (saliva) and then detect by the proposed assay. In brief, 30 μL of running buffer (10 mM PBS, 3% Tween, and 10% FBS) was added into 60 μL of the tested samples containing various concentrations of *S. pneumoniae*, the mixture was vigorously vortexed for 10 s and then pipetted onto the sample pad of ICA strip. After 15 min of the chromatographic reaction, the colorimetric and SERS signal intensities of test lines were measured by the naked eye and Raman spectrometer, respectively, for the direct and quantitative detection of bacteria.

## 3. Results and Discussion

### 3.1. Preparation and Characterization of SiO_2_@20Au SERS Tags

The SiO_2_@20Au core-shell SERS tags were synthesized via a PEI-mediated electrostatic adsorption approach as illustrated in Figure 1A, which consisted of four parts: (i) a 180 nm SiO_2_ nanosphere as a stable and monodispersed supporter to guarantee good dispersion and high stability in complex biological samples; (ii) a layer of positively charged PEI shells to act as a bridge connecting SiO_2_ to 20 nm AuNPs; (iii) a layer of dense AuNPs formed shell to generate strong SERS activity and numerous surface sites for bacteria detection; (iv) a layer of DTNB molecules to provide specific Raman signals for target quantitation and abundant surface carboxyl groups for antibody conjugation. 

The surface morphology and structural composition of SiO_2_@20Au NPs were characterized by transmission electron microscopy (TEM) and energy-dispersive X-ray spectroscopy (EDS) elemental analysis. Figure 1A,B,D,E display the typical TEM images of SiO_2_, SiO_2_@PEI, 20 nm AuNPs, and SiO_2_@20Au NPs, respectively. Obviously, the fabricated 180 nm SiO_2_ presented a spherical morphology with uniform particle size and a smooth surface. Our previous works have proven that PEI can self-assemble on SiO_2_ surfaces to form a positively charged thin shell under ultrasonic conditions [37,38]. High-resolution TEM was used to verify the PEI coating, and the image in Figure 1C clearly shows the thickness of the PEI layer is around 2 nm. Herein, negatively charged AuNPs with a suitable size (20 nm) were used as satellites to fabricate the SiO_2_@20Au core-shell nanocomposites (Figure 1D). As shown in Figure 1E, numerous 20 nm AuNPs can be firmly adhered onto the surface of SiO_2_@PEI surface through the strong electrostatic adsorption of the PEI shell, thus forming a typical core-satellite structure. The EDS elemental line scan result (Figure 1F) and EDS elemental mapping (Figure 1G) result for a single SiO_2_@20Au NP revealed that the outer layer of dense AuNPs (blue) were uniformly distributed on the surface of the SiO_2_ core (red and green), which clearly demonstrated the structural components of the proposed SERS tag. In addition, the amount of AuNPs loaded onto SiO_2_ can be determined by EDS spectroscopy. As revealed in Appendix A, the EDS spectrum indicates the presence of obvious Si, O, and Au signals in the SiO_2_@20Au nanostructure with the corresponding elemental composition (atomic fraction) of 31.04%, 55.80%, and 13.17%, respectively. The UV-vis spectra of SiO_2_@20Au NPs showed an obvious absorption peak at 531 nm after the coating of 20 nm AuNPs, indicating the strong coupling of dense AuNPs formed shells (Appendix A). All the above results confirmed the successful fabrication of SiO_2_@20Au NPs. In addition, the zeta potential values in Figure 1H reveal the surface potential of the nanomaterials in each stage. The zeta potential of SiO_2_ NPs increased markedly after PEI coating and decreased sharply after the adsorption of AuNPs, suggesting the assembly of SiO_2_@20Au core-shell nanocomposites was based on electrostatic interaction.

Given the numerous AuNPs coated onto the SiO_2_ surface and the hotspots created on the gaps between the two adjacent AuNPs, the SiO_2_@20Au composites exhibited superior SERS activity than the common AuNPs. Figure 2A displays the SERS spectra of the Raman reporter molecule DTNB adsorbed onto the prepared nanomaterials. The SERS spectra were baseline subtracted and shifted vertically for visualization. The SiO_2_ and SiO_2_@PEI NPs had no enhancement effect (black line and red line) and the SiO_2_@20Au showed the best enhancement on DTNB (green line). By comparing the main peak of DTNB (1331 cm^−1^), the SERS performance of SiO_2_@20Au was increased about 2 times that of colloidal AuNPs. We next assessed the colloidal stability of SiO_2_@20Au in complex environments through comparison with the common AuNPs. The NP aggregation in complex samples is the main cause of nonspecific signals of label-based SERS immunoassays. As shown in Figure 2B, the SiO_2_@20Au-DTNB exhibited rather good dispersibility (i) and stable SERS signals (ii) in high salt samples (100–1000 mM), whereas the colloidal AuNPs were severely agglomerated in 100 mM NaCl solution. Moreover, the prepared SiO_2_@20Au-DTNB exhibited excellent colloidal stability and strong SERS intensity in an aqueous solution at different pH values (3–13) (Figure 2C). These results confirmed that the large SiO_2_ core greatly improved the stability of SiO_2_@20Au in complex samples.

### 3.2. Design and Optimization of SERS-ICA for S. pneumoniae Detection

The SERS-ICA system for sensitive detection of *S. pneumoniae* was designed as shown in Figure 1C, which consisted of a liquid SiO_2_@20Au SERS tag, a sample pad for sample loading, an NC membrane with one test line for target bacteria capturing, and an absorption pad to generate capillary force. The testing process of the proposed assay could be completed in two simple steps. Firstly, the liquid SiO_2_@20Au SERS tags were mixed with the sample solution to ensure full immunobinding to target bacteria. Secondly, the bacteria/SERS tags mixture was loaded onto the sample pad of the ICA strip to start the chromatographic reaction. Driven by capillary action force, the solution moved forward to the test line, and the *S. pneumoniae*/SiO_2_@20Au SERS tag immunocomplexes will bind to the capture antibody on the test zone to form sandwich complexes. The excess SiO_2_@20Au SERS tags continued to flow forward to the control line and were caught by the goat anti-mouse IgG, which can generate a black band for quality control of the ICA strip. In theory, higher concentrations of *S. pneumoniae* existed in tested samples, more bacteria/SiO_2_@20Au complexes were immobilized onto the T line, and a stronger SERS intensity was produced in the test zone. The detailed SERS signals of the test lines can be rapidly measured by using a portable Raman spectrometer and used for the quantitative analysis of *S. pneumoniae.* Notably, 785 nm laser excitation was chosen because it is suitable to reduce the fluorescence background of the NC membrane and the damage to biological samples [29,39].

To achieve the best performance on the SiO_2_@20Au-based SERS-ICA system, the optimal conditions of SERS tags and ICA strip were studied. The anti-bacterial antibody can be easily conjugated onto the surface of SiO_2_@20Au-DTNB via EDC/NHS activation, due to the terminal carboxyl group of DTNB [40]. Zeta potential, dynamic light scattering (DLS) and Fourier transform infrared spectroscopy (FTIR) were used to evaluate the antibody modification effect on the SiO_2_@20Au surface. As revealed in Figure 3A, the zeta potential values of SiO_2_@20Au-DTNB decreased with an increase in the antibody dosage in the reaction system (0–8 μg) and remained stable at −28.4 mV, indicating the antibody amount modified onto SERS tags has reached saturation. In addition, the DLS (Appendix A) and FTIR (Appendix A) results in the Appendix A also verified the successful conjugation of anti-*S. pneumoniae* antibody onto SiO_2_@20Au surface. The binding ability of antibody-modified SiO_2_@20Au SERS tags toward their target bacteria was next assessed. The standard strain of *S. pneumoniae* was supplied by Prof. Bing from Guangdong Provincial People’s Hospital and verified by PCR (Figure 3B). The prepared *S. pneumoniae* was incubated with antibody-modified SiO_2_@20Au SERS tags and then investigated by TEM. As shown in Figure 3C,D, the immuno-SiO_2_@20Au tags can effectively bind to the surface of *S. pneumoniae*, suggesting the high affinity of the used antibody to the target bacteria. 

The key conditions of ICA strips mainly included the pore size of the NC membrane, running buffer composition, and chromatographic time. Our previous works have demonstrated that the CN95 NC membrane with a 15 µm pore size can provide enough wide transport channels for large nanotags (>200 nm) and is suitable for bacteria detection [41,42,43]. With the CN95 membrane, in this work, we tested several widely used running buffers to ensure the good mobility and analytical performance of the SERS-ICA for bacteria. As shown in Appendix A, the PBS buffer containing 1% Tween 20 and 5% FBS produced the strongest SERS signals and the best signal-to-noise ratio (SNR) for *S. pneumoniae* detection. In addition, the chromatographic time of SiO_2_@20Au-based ICA was investigated, as shown in Appendix A. The optimization results indicated that the 15 min chromatographic time can generate the best testing results on ICA strips.

### 3.3. Detection Performance of SiO_2_@20Au-SERS-ICA for S. pneumoniae

To evaluate the ability of the proposed assay for direct detection of bacteria, the SiO_2_@20Au-SERS-ICA system was used to analyze a series of bacterial samples containing various concentrations (10^6^–0 cells/mL) of *S. pneumoniae.* The photos of the tested ICA and the SERS mapping results corresponding to the test lines (T lines) are shown in Figure 4A. Obviously, the dark colors on the T lines were gradually weakened with the decrease in the concentration of *S. pneumoniae* in the sample solution. The T line was still observable at 1 × 10^3^ cells/mL of *S. pneumoniae* with the naked eye, indicating the visual sensitivity of the proposed SERS-ICA was about 1 × 10^3^ cells/mL. In addition, the SERS mapping images (22 × 8 pixels), which were obtained from the test zones using the specific signals of DTNB (1331 cm^−1^) as a source, exhibited the relatively uniform SERS signal intensity distributed on the T lines when *S. pneumoniae* concentration in the sample is higher than 500 cells/mL. To achieve accurate detection, 20 spots on the one T line were randomly measured and averaged to generate a reproducible SERS signal. The SERS spectrum of different concentrations of *S. pneumoniae* was shown in Figure 4C. Notably, a rather weak SERS signal (~1331 cm^−1^) on the test line can be still detected when the *S. pneumoniae* concentration is 0 cells/mL. This weak SERS signal is derived from the residual SERS tags in the running solution and can be deducted as the background signal. The calibration curve for *S. pneumoniae* was constructed by utilizing the sigmoid function of bacteria concentration and the corresponding SERS signals at 1331 cm^−1^, as shown in Figure 4D. The limit of detection (LOD) of the proposed SiO_2_@20Au-SERS-ICA for *S. pneumoniae* was calculated using the IUPAC protocol (LOD = y_blank_ + 3 SD_blank_), which was determined to be 46 cells/mL. The detection range of the SERS-ICA for *S. pneumoniae* spanned five orders of magnitude (10^5^–50 cells/mL) with correlation coefficients (R^2^) = 0.995. Moreover, the limit of quantitation (LOQ) of SiO_2_@20Au-ICA was found to be 64 cells/mL, according to ten times the standard deviation of the blank control. The superiority of SiO_2_@20Au-based SERS-ICA was further intuitively evaluated by using the commonly used AuNP-based colorimetric ICA method as a comparison. The AuNP-ICA strip for *S. pneumoniae* detection was prepared using the same immunoreagents with SiO_2_@20Au-ICA and the detailed preparation process was shown in Appendix A. As shown in Figure 4B, the visual sensitivity levels of the AuNP-based ICA method for *S. pneumoniae* observed with the naked eye was 5 × 10^3^ cells/mL. Thus, the LOD of the AuNP-ICA strips can be determined to be 5 × 10^3^ cells/mL by the colorimetric signal. By comparison, the SiO_2_@20Au-ICA strip based on the SERS signal can achieve about 100 times improvement in sensitivity for *S. pneumoniae* detection than traditional AuNP-based ICA. 

### 3.4. Reproducibility and Specificity of SiO_2_@20Au-SERS-ICA 

As a potential POCT reagent for the rapid detection of respiratory pathogens, the stability and reproducibility of SiO_2_@20Au-SERS-ICA need to be assessed. We prepared three groups of *S. pneumoniae* samples at a high concentration (10^6^ cells/mL), medium concentration (10^4^ cells/mL), and low concentration (10^2^ cells/mL) and used them for performance testing. The photographs of tested ICA strips and the corresponding SERS signals on the T lines from five independent experiments are shown in Figure 5A(i),(ii), respectively. From these results, we found that the SiO_2_@20Au-SERS-ICA exhibited excellent reproducibility of SERS signals for all the tested groups. The relative standard deviations (RSD) values for different concentrations (10^6^, 10^4^, and 10^2^ cells/mL) of target bacteria were less than 6.8%, which clearly demonstrated the good stability of our ICA method. In addition, the specificity of SiO_2_@20Au-SERS-ICA was verified by detecting other common pathogenic bacteria including *Staphylococcus aureus* (*S. aureus*), *Escherichia coli* (*E. coli*), *Salmonella typhimurium* (*S. typhimurium*), *Listeria monocytogenes* (*L. mono*), and *Staphylococcus epidermidis* (*S. epidermidis*) and common respiratory viruses including influenza A virus (Flu A) and influenza B virus (Flu B). As revealed in Figure 5B, only the samples containing *S. pneumoniae* (10^4^ cells/mL) can be recognized by the proposed ICA, resulting in a distinct dark band and strong SERS signal on the test line. All the non-target pathogens could not generate obvious SERS signals on the T lines of ICA strips, indicating the superior specificity of our method.

### 3.5. Detection in Spiked Biological Samples 

We finally investigated the performance of the SiO_2_@20Au-ICA strip in real biological samples. The human sputum samples collected from healthy volunteers were thoroughly sterilized through thermal processes and spiked with *S. pneumoniae* at concentrations of 10^4^, 10^3^, and 10^2^ cells/mL. The precision of our SERS-ICA was determined by the recovery test of these samples. The obtained SERS spectra for each sample were averaged to generate a reproducible signal (Appendix A) and substituted into the established calibration curves to calculate the recoveries of *S. pneumoniae* in the real sputum samples. As summarized in Appendix A, the recovery rates of SiO_2_@20Au-ICA for spiked *S. pneumoniae* were calculated to be 89.2–124.6%, with RSD values below 11.3%. These results confirmed the good accuracy of the proposed SERS-ICA for the complex biological samples. The excellent detection performance (sensitivity, stability, specificity, and accuracy) of the proposed SERS-ICA can be attributed to the superior properties of SiO_2_@20Au tags, including high SERS activity, good dispersibility in complex samples, and numerous surface-active sites for bacteria binding. In theory, by using the specific antibodies for target pathogens, the proposed SERS-ICA method can be easily used to detect other pathogenic microorganisms and has great potential to be developed further.

## 4. Conclusions

We proposed a simple SERS-ICA method for rapid and sensitive detection of *S. pneumoniae* in complex samples by using a high-performance SiO_2_@20Au SERS tag. The SiO_2_@20Au tag was prepared by one-step adsorption of numerous 20 nm AuNPs onto 180 nm SiO_2_ NPs and then introduced into the ICA strip to replace conventional colloidal SERS tags. The SiO_2_@20Au tag can provide stronger SERS activity, better dispersibility, higher stability, and larger surface sites than the commonly used AuNP SERS tag. The proposed SiO_2_@20Au-SERS-ICA achieved the rapid detection of *S. pneumoniae* in 20 min with an LOD value of 46 cells/mL. Remarkably, the sensitivity of our proposed ICA was over 100 times higher than that of traditional AuNP-based ICA strips. Based on these findings, we believe that the proposed assay has great potential in the POCT detection of pathogens.

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
