# Peer review of "Electrostatic Adsorption of Dense AuNPs onto Silica Core as High-Performance SERS Tag for Sensitive Immunochromatographic Detection of *Streptococcus pneumoniae"

_pathogens, 2023, doi:10.3390/pathogens12020327_

Round 1

Reviewer 1 Report

This is a very straightforward paper, with carefully presented results. There is some lack of characterization of the particles and conjugates of particles, that assure the proposed system performance, such as, size distribution (average +/- SD), AuNPs per SiO2NP +/- SD, etc. 

Minor comments: 

Title: The authors call these AuNPs “dense AuNPs - why dense? 

Line 109:50 mL of 20 nm AuNPs. The 20 mL are of a suspension. What is the concentration of this suspension? If the authors do not have this information, please inform if the suspension is used as synthesized, or if there are concentration/dilution steps, resulting from washing, for example. 

Lines 114, 121: Please check “S. pneumoniae” formatting. 

Line 126: Please check the typo in NaN3 formula. 

Line 130: Please define NC as nitrocellulose 

Line 181: “Transmission”: No need to use a capital T. Why is TEM defined and EDS is not? 

Line 218: NPs aggregation is not necessarily bad for SERS. If it is bad for your assembling, you should differentiate these two aspects. 

Lines 263 and others (sections 3.3, 3.4, 3.5, etc.): Text not formatted according to the template (font). 

Author Response

Dear Editor and Reviewers,

Thank you for your letter and for the reviewer’s comments concerning our manuscript entitled “Electrostatic adsorption of dense AuNPs onto silica core as high-performance SERS tag for sensitive immunochromatographic detection of Streptococcus pneumoniae”. Those comments are valuable and very helpful for revising and improving our manuscript. We have studied the comments carefully and have made corrections which we hope to meet the reviewer’s request. The revised portions are marked in red in the paper. The main corrections in the paper and the response to questions brought up by the academic editor and reviewers are listed as below:

To referee 1

Comments:

This is a very straightforward paper, with carefully presented results. There is some lack of characterization of the particles and conjugates of particles, that assure the proposed system performance, such as, size distribution (average +/- SD), AuNPs per SiO2NP +/- SD, etc.

Reply: We appreciate reviewer 1 for the valuable comments and the time to review our manuscript. We have read the comments carefully and it is really helpful to improve the level of our research paper. We have tried our best to revise the manuscript. Would you please spend time to review the revision? Thanks so much for your kindly help.

Minor comments:

  1. Title: The authors call these AuNPs “dense AuNPs” - why “dense”?

Answer: Thank you very much for your careful review and kind advice. In this study, the SiO2@20Au tag was prepared by using an excessive amount of 20 nm AuNP solution. As we described in Experimental section 2.2, the PEI-coated SiO2 NPs were mixed with excess AuNP (20 nm) solution under sonication and quickly absorbed many AuNPs onto their surface by electrostatic interaction. Notably, in the theory of SERS enhancement, the denser of AuNPs onto the surface of supporter, the stronger SERS intensity (Gandra et al., Nano Lett. 2012, 12, 2645-2651; Ye et al., Nanoscale. 2015, 7(32), 13427-37; Cheng et al., Nanoscale, 2015, 7, 3445; Wang et al., Biosensors and Bioelectronics 214 (2022) 114525). To achieve the fully grafting of AuNPs onto SiO2 surface, the sonication times were optimized. As shown in Fig. R1, the number of AuNPs attached onto the surface of SiO2-PEI surface increased with the sonication time (0-30 min) and reached saturation at 30 min. Moreover, the SERS intensity values of SiO2@20Au-DTNB NPs increased with increasing amount of AuNPs. From these results, the dense AuNPs onto SiO2 NPs is the key factor in the SERS performance of SiO2@20Au tag. Thus, we want to emphasize this point (dense AuNPs) in the title of the article.

Fig. R1 (A-D) TEM images of fabricated SiO2@20Au NPs with different sonication times of AuNP solution (0-30 min). (E) SERS spectra and (F) corresponding SERS intensity (1331 cm-1) of SiO2@20Au-DTNB with different AuNPs amount under the same condition.

  1. Line 109: “50 mL of 20 nm AuNPs”. The 20 mL are of a suspension. What is the concentration of this suspension? If the authors do not have this information, please inform if the suspension is used as synthesized, or if there are concentration/dilution steps, resulting from washing, for example.

Answer: Thank you very much for your careful review and professional suggestion. In this work, the 20 nm AuNPs were fabricated using the typical trisodium citrate reduction method (Nat Protoc. 2013, 8(1), 52-65). As we described in the Experimental section, 1 mL of 1% (W/V) HAuCl4 solution was added into 99 mL of deionized water to prepare Au NPs.

Thus, the Au concentration is 0.01% = 2.94´10-4 mol/L

If we centrifuge 1 mL, it contains 1.0 ml ´ 2.94 ´ 10-4 mol/L = 2.94´10-7 mol Au

A 20 nm Au NPs volume = (4/3)´3.14´(20 nm/2)3 = 4186.66 nm3 = 4186.66 ´10-28 m3

The molar volume for Au is 10.21 cm3/mol, so a 20 nm Au NP has 4186.66 ´10-28 m3 / 10.21 cm3/mol = 410.05´10-22 mol

So 1.0 mL Au solution has Au NP numbers = 2.94´10-7 mol / 410.05´10-22 mol = 7.17´1012 particles

In addition, many previous literatures have been shown that the concentration of 20 nm Au NPs produced by this method is generally 10 nM (For example, Biosensors and Bioelectronics 78 (2016) 80–86; Sensors and Actuators B: Chemical 262, 733-738; ACS Nano. 2010, 4, 7451-8). Notably, the AuNPs (20 nm) suspension has not been concentrated or diluted before use. The SiO2@20Au SERS tags were prepared by electrostatic adsorption of dense 20 nm AuNPs onto 180 nm SiO2 core. To ensure the fully adsorption of AuNPs onto SiO2 core, excessive amount of 20 nm AuNPs (50 mL) were used. The resulting SiO2@20Au NPs were separated by centrifugation (4500 rpm, 6 min) to remove the excess AuNPs, and the precipitate was redispersed and stored in 10 mL ethanol for later use.

We appreciate the referee’s suggestion very much and have added a more detailed explanation on the AuNPs concentration to make our manuscript more rigorous. Please see the red words in the Experimental section 2.2 (Page 6).

  1. Lines 114, 121: Please check “S. pneumoniae” formatting.

Answer: Thank you very much for your careful review and kind advice. The “S. pneumoniae” formatting has been revised.

  1. Line 126: Please check the typo in NaN3 formula.

Answer: Thank you very much for your careful review and kind advice. We are very sorry for this mistake, and the NaN3 formula has been corrected in the revised manuscript.

  1. Line 130: Please define NC as nitrocellulose.

Answer: Thank you very much for your careful review and kind advice. The full name of NC (nitrocellulose) membrane has been added in the revised manuscript.

  1. Line 181: “Transmission”: No need to use a capital T. Why is TEM defined and EDS is not?

Answer: Thank you very much for your careful review and kind advice. The correction of “transmission” has been made as you suggested. In addition, the full name of EDS (energy-dispersive X-ray spectroscopy) has been added in the revised manuscript.

  1. Line 218: NPs aggregation is not necessarily bad for SERS. If it is bad for your assembling, you should differentiate these two aspects.

Answer: Thank you very much for your careful review and professional suggestion. It is indeed that NPs aggregation may increase the sensitivity of label-free SERS detection (using Raman fingerprint). However, the NPs aggregation is the main cause of nonspecific signal of label-based SERS immunoassays (Chem. Rev. 2013, 113, 1391-1428; Chem. Mater. 2015, 27, 950-958; J. Mater. Chem. C, 2020,8, 2142-2154; Chemical Engineering Journal 448 (2022) 137760; Anal. Chem. 2017, 89, 10104-10110).

We appreciate the referee’s suggestion very much, and we have added more content to explain this issue in the revised manuscript to make our work much clearer to the readers (Page 11).

  1. Lines 263 and others (sections 3.3, 3.4, 3.5, etc.): Text not formatted according to the template (font).

Answer: Thank you very much for your careful review and kind advice. The correction has been made as you suggested.

We appreciate for Editor/Reviewers’ warm work earnestly, and hope that the correction will meet with approval. Once again, thank you very much for your comments and suggestions.

Reviewer 2 Report

The manuscript pathogens-2192852 proposed a simple SERS-ICA method for rapid and sensitive detection of S. pneumoniae in complex samples. The work is fundamentally sound and I found no fault whatsoever with the data analysis, or conclusions. The manuscript is clear and well-written (just a few typos) with a potential for publication. My comments here are mainly aimed at tightening up the logic and clarity of what was communicated and tested:

- If the developed method aims at reproducible quantitative detection, the calibration curve (Figure 4D) must be improved with confidence and prediction bands. Error bars expressed for each calibration level (are they standard deviations or confidence intervals? how many replicates?) are not worth very much in a calibration plot. They can be useful to check the homogeneity of the variance. Another point is the non linear logistic regression used as a calibration curve. Why? Since the authors used IUPAC definition of LOD I wonder why not use the proper calibration practice as suggested by IUPAC (https://doi.org/10.1351/pac199870040993). Please comment. For the same reason please also provide the experimental Limit of quantitation (LOQ). It is a very important Figure of merit in quantitative application since the linearity range for the sensor should properly begin at the LOQ level. 

-Dynamite plots (Figures 1H, 2Cii, 5Bii) are never an appropriate way to plot the data. Even if they are still regularly used, they are universally considered a “bad practice”, well documented in the literature (see, for instance, 10.1111/j.1476-5381.2011.01251.x).  If the authors still want to preserve the dynamite plot visualization, they should at least specify in the caption the meaning of the error bars (are they a confidence interval or a standard deviation?) and the number of replicates.

- It is not clear if the SERS spectra in Figure 5 are mean spectra from a SERS map or if they are single spectra. Please clarify it.

- The authors declared (Conclusions) that "the sensitivity of our proposed ICA was over 100 times 356 higher than that of traditional AuNP-based ICA strip", however, the exact value of sensitivity for the proposed method was never mentioned. Only visual sensitivity (from naked eye) was reported. Similarly, accuracy was mentioned several times but never quantitatively reported. Please report sensitivity and accuracy for the developed SERS method.

Author Response

Dear Editor and Reviewers,

Thank you for your letter and for the reviewer’s comments concerning our manuscript entitled “Electrostatic adsorption of dense AuNPs onto silica core as high-performance SERS tag for sensitive immunochromatographic detection of Streptococcus pneumoniae”. Those comments are valuable and very helpful for revising and improving our manuscript. We have studied the comments carefully and have made corrections which we hope to meet the reviewer’s request. The revised portions are marked in red in the paper. The main corrections in the paper and the response to questions brought up by the academic editor and reviewers are listed as below:

To referee 2

Comments:

The manuscript pathogens-2192852 proposed a simple SERS-ICA method for rapid and sensitive detection of S. pneumoniae in complex samples. The work is fundamentally sound and I found no fault whatsoever with the data analysis, or conclusions. The manuscript is clear and well-written (just a few typos) with a potential for publication. My comments here are mainly aimed at tightening up the logic and clarity of what was communicated and tested.

Reply: We appreciate reviewer 2 for the valuable comments and the time to review our manuscript. We have read the comments carefully and it is really helpful to improve the level of our research paper. We have tried our best to revise the manuscript. Would you please spend time to review the revision? Thanks so much for your kindly help.

Questions:

  1. If the developed method aims at reproducible quantitative detection, the calibration curve (Figure 4D) must be improved with confidence and prediction bands. Error bars expressed for each calibration level (are they standard deviations or confidence intervals? how many replicates?) are not worth very much in a calibration plot. They can be useful to check the homogeneity of the variance.

Answer: Thank you very much for your careful review and professional suggestion. In this work, to achieve the rapid detection of the Raman signal on the ICA strip, a portable Raman spectrometer (B & W Tek, i-Raman Plus BWS465-785H) equipped with a 785 nm laser was utilized. As we described in Supporting information S1.1 Instruments and SERS measurements conditions, the SERS measurement on the ICA strips was conducted with the following conditions: 5 s exposure time, 5 mW laser power, and ~100 μm spot size. To reduce experimental error, twenty random tests were randomly measured on the test zone for each sample and the obtained SERS spectra were averaged for subsequent analysis. For validation of the result, different groups of samples were measured independently and all the experiments were performed three times. According to the previous report, sampling errors can be curtailed by increasing the sample size (Anal. Chem., 2018, 90 (2), 1248–1254). Our previous works demonstrated that the established SERS detection protocol works well in the SERS–ICA biosensors (Wang et al., Biosensors and Bioelectronics 214 (2022) 114525; Liu et al., Sens Actuators B Chem. 2021, 329, 129196; Liu et al., Sens. Actuators B-Chem., 2020, 320, 128350; Wang et al., ACS Appl. Mater. Interfaces. 2019, 11 (21), 19495).

The Figure 4c and Figure 4d in this work showed the SERS spectra of tested strips that corresponding to different concentrations of S. pneumoniae and the corresponding calibration curve, respectively. Error bars represent the standard deviations from three repetitive experiments. We appreciate the referee’s suggestion and have added a more detailed description of experiment condition and error bars in the revised manuscript (page 15) and supporting information S1.1.

  1. Another point is the non linear logistic regression used as a calibration curve. Why? Since the authors used IUPAC definition of LOD I wonder why not use the proper calibration practice as suggested by IUPAC (https://doi.org/10.1351/pac199870040993). Please comment.

Answer: Thank you very much for your careful review and professional suggestion. In this manuscript, the readouts against S. pneumoniae concentration were fitted by a sigmoidal dose−response curve with a four-parameter logistic equation. The plot for S. pneumoniae exhibited logarithmic relationship with small standard deviations over a wide detection range (105–10 cells/mL). Moreover, the readouts showed strong correlation up to 0.995. The LOD was calculated from the sigmoidal curve fit as the S. pneumoniae concentration intersecting a line representing the intensity value that exceeds three times the standard deviation (SD) of the blank measurements (the standard IUPAC method). The IUPAC method is widely used for ICA strip-based detection, such as “Liu et al., Sensors and Actuators: B. Chemical 329 (2021) 129196; Hu et al, Anal. Chem. 2017, 89, 13105−13111; Wang et al, Anal. Chem. 2017, 89, 1163−1169; Fu et al, Biosensors and Bioelectronics 78 (2016) 530–537; Cheng et al., Chemical Engineering Journal 426 (2021) 131836”.

The LOD can be calculated with the following formula:

LOD = yblank + 3 × SDblank, where yblank and SDblank are the average SERS intensity and standard deviation of the blank control, respectively.

Using this equation, the LOD of S. pneumoniae was calculated to be 46 cells/mL.

Other fitting calibration curves, such as linear fit, polynomial fit or exponential fit, can’t provide a similar correlation.

Notably, the sigmoidal curve fit is widely used in ICA strip-based detection method. For example, “Wang et al, Anal. Chem. 2017, 89, 1163−1169; Brangel et al, ACS Nano 2018, 12, 63−73; Wang et al, Sensors and Actuators B 270 (2018) 72–79; Rong et al, Analyst, 2018, 143, 2115; Rong et al, Analytica Chimica Acta 1055 (2019) 140e147; Liu et al., Sensors & Actuators: B. Chemical 320 (2020) 128350”. Therefore, the sigmoidal curve fit used in this study is reasonable.

  1. For the same reason please also provide the experimental Limit of quantitation (LOQ). It is a very important Figure of merit in quantitative application since the linearity range for the sensor should properly begin at the LOQ level.

Answer: Thank you very much for your careful review and professional suggestion. The LOQ of the SiO2@20Au-ICA strip for S. pneumoniae has been added in the revised manuscript (Page 14).

  1. Dynamite plots (Figures 1H, 2Cii, 5Bii) are never an appropriate way to plot the data. Even if they are still regularly used, they are universally considered a “bad practice”, well documented in the literature (see, for instance, 10.1111/j.1476-5381.2011.01251.x). If the authors still want to preserve the dynamite plot visualization, they should at least specify in the caption the meaning of the error bars (are they a confidence interval or a standard deviation?) and the number of replicates.

Answer: Thank you very much for your careful review and kind advice. Histograms are widely used in SERS-ICA based methods which could easy and intuitive to show the results. So we want to keep these histograms in this work. We appreciate the referee’s suggestion very much and have added the meaning of the error bars and the number of replicates in the caption of the Figures in the revised manuscript.

  1. It is not clear if the SERS spectra in Figure 5 are mean spectra from a SERS map or if they are single spectra. Please clarify it.

Answer: Thank you very much for your careful review and kind advice. The SERS spectra in Figure 5 is the mean spectrum of twenty SERS single spectra measured from the test line. The related content has been added in the caption of the Figure 5 to make our manuscript much clearer to our readers.

  1. The authors declared (Conclusions) that "the sensitivity of our proposed ICA was over 100 times 356 higher than that of traditional AuNP-based ICA strip", however, the exact value of sensitivity for the proposed method was never mentioned. Only visual sensitivity (from naked eye) was reported. Similarly, accuracy was mentioned several times but never quantitatively reported. Please report sensitivity and accuracy for the developed SERS method.

Answer: Thank you very much for your careful review and kind advice. Technically speaking, the SERS-ICA strip for target detection is an indirect detection, and the signal is reported by the DTNB-modified SiO2@20Au SERS tags. When the tested samples contained the target bacteria, the SiO2@20Au tags will bind to the bacterial surface and the formed SiO2@20Au-bacteria complexes were specifically identified and captured by the anti-S. pneumoniae antibody on the test line. The overall intensity distribution of the SERS signals on the test line increased as the target concentration gradually increased. Many previous works have demonstrated that this phenomenon can be utilized for the quantitative evaluation of the target (For example, Anal. Chem. 2021, 93, 8362-8369; ACS Appl. Mater. Interfaces 2019, 11, 19495-19505; Angew Chem Int Ed Engl. 2019, 58(2), 442-446; Biosensors and Bioelectronics 106 (2018) 204–211; , Biosensors and Bioelectronics 214 (2022) 114525). In this work, the LOD of our method is calculated as the corresponding concentration to the value: yblank + 3 × SDblank, where yblank is the average value and SDblank is the standard deviation of the background SERS peak intensity at 1331 cm−1, based on the calibration curve in Figure 4d. Thus, the LOD of our SERS-ICA strip was estimated to be 46 cells/mL by using SERS signal.

To directly compare the sensitivity of the proposed SiO2@20Au-ICA with the traditional ICA methods, the standard AuNP-based ICA strip for S. pneumoniae detection was carried out. As shown in Fig. 4B, the visual sensitivity levels of the AuNP-based ICA method for S. pneumoniae observed with the naked eye was 5 × 103 cells/mL. Thus, the LOD of the AuNP-ICA strips can be determined to be 5 × 103 cells/mL by the colorimetric signal. By comparison, the SiO2@20Au-ICA strip based on SERS signal can achieve about 100 times improvement in sensitivity for S. pneumoniae detection than traditional AuNP-based ICA. Notably, this comparison way has been wildly used in novel signal tags-based ICA methods (For example, Wang et al., ACS Appl. Mater. Interfaces. 2019, 11 (21), 19495; Cheng et al., Chemical Engineering Journal 426 (2021) 131836; Zheng et al., Sensors and Actuators: B. Chemical 359 (2022) 131528; Wang et al., Anal. Chem. 2020, 92, 15542-15549; Zheng et al., Chemical Engineering Journal 448 (2022) 137760; Shen et al., Journal of Hazardous Materials 437 (2022) 129347).

In addition, the accuracy of the current ICA methods was generally verified through the recovery studies (For example, Wang et al., Nanoscale, 2020, 12, 795–807; Qie et al., Anal. Chem. 2019, 91, 9530-9537; Sheng et al., Biosensors and Bioelectronics 181 (2021) 113149; Liu et al., Anal. Chem. 2021, 93, 3626-3634; Zheng et al., Food Chemistry 363 (2021) 130400). As summarized in Table S1, the recovery rates of SiO2@20Au-ICA for spiked S. pneumoniae were calculated to be 89.2-124.6 %, with RSD values below 11.3 %. These results confirmed the good accuracy of proposed SERS-ICA for the complex biological samples.

Thus, we think the sensitivity and accuracy of the proposed SERS-ICA method has been carefully investigated in the manuscript under the universal standard of current ICA technologies.

We appreciate for Editor/Reviewers’ warm work earnestly, and hope that the correction will meet with approval. Once again, thank you very much for your comments and suggestions.

Reviewer 3 Report

1. Firstly, this work present a really low novelty.

Authors describe recently: 

2022 August: Magnetic Nanotag-Based Colorimetric/SERS Dual-Readout Immunochromatography for Ultrasensitive Detection of Clenbuterol Hydrochloride and Ractopamine in Food Samples (Same NPs and same ICA for detection of antibiotics).

2022 August: Nanogapped Fe3O4@Au Surface-Enhanced Raman Scattering Tags for the Multiplex Detection of Bacteria on an Immunochromatographic Strip (Diferent NPs for bacteria detection).

2022 September: Ultrasensitive multichannel immunochromatographic assay for rapid detection of foodborne bacteria based on two-dimensional film-like SERS labels (Similar technology for Bacteria detection)

Even more, figures are very similar. 

2. Authors must describe organic layer in deep. FTIR, TGA and DLS to check that antibodies are properly linked and yield.... 

3. Authors describe "obvious absorption peak at 531 nm". Since SERS is optimum if SPR of NPs is close to laser wavelength, why authors choose these gold NPs?

4. Did authors take into account that y axis should not have units? If authors consider that units must be showed, Why some spectra start in 10000?

5. In order to know the amount of gold which is linked to SiO2, authors must perform an ICP analysis.

6. Figure 3c. Did authors used Cryo-TEM to see bacteria?? Metal stains?

7. Figure 4c. Could authors explain why SERS spectrum of 0 cells/ml have the main peaks of bacteria, but with less intensity? Moreover, Figure 4c has 9 spectra, while Figure 4d only have into account 7 intensities. Please, explain it.

8. Most important issue. Section 3.5 describes real samples. Sputum with the addition in the laboratory of bacteria are not real sample. Authors must evaluate real patient samples in order to improve the novelty of the work. Moreover, authors must show not only a table in the ESI, but all SERS spectra in a Figure in the main text. 

Author Response

Dear Editor and Reviewers,

Thank you for your letter and for the reviewer’s comments concerning our manuscript entitled “Electrostatic adsorption of dense AuNPs onto silica core as high-performance SERS tag for sensitive immunochromatographic detection of Streptococcus pneumoniae”. Those comments are valuable and very helpful for revising and improving our manuscript. We have studied the comments carefully and have made corrections which we hope to meet the reviewer’s request. The revised portions are marked in red in the paper. The main corrections in the paper and the response to questions brought up by the academic editor and reviewers are listed as below:

To referee 3

Comments:

  1. Firstly, this work present a really low novelty. Authors describe recently:

2022 August: Magnetic Nanotag-Based Colorimetric/SERS Dual-Readout Immunochromatography for Ultrasensitive Detection of Clenbuterol Hydrochloride and Ractopamine in Food Samples (Same NPs and same ICA for detection of antibiotics).

2022 August: Nanogapped Fe3O4@Au Surface-Enhanced Raman Scattering Tags for the Multiplex Detection of Bacteria on an Immunochromatographic Strip (Diferent NPs for bacteria detection).

2022 September: Ultrasensitive multichannel immunochromatographic assay for rapid detection of foodborne bacteria based on two-dimensional film-like SERS labels (Similar technology for Bacteria detection)

Even more, figures are very similar.

Answer: Thank you very much for your careful review and kind advice. We really appreciate your careful review and good suggestions for our improved manuscript. According to your comments, we have carefully revised our paper and made some changes. Please see the details below.

In recent years, the SERS-based ICA method has been considered as one of the most sensitive paper-based POCT techniques, because it uses the Raman dye-modified SERS tags to provide ultra-sensitive (single-molecule-level), specific (characteristic peak), and stable (no photobleaching) Raman signal for target determination. Though many SERS tags have been proposed and integrated into ICA system, several major limitations of SERS-LFA methods in practical applications for complex sample analysis remain to be improved.

First, although the SERS tag is the key determining the performance of SERS-LFA, the commonly used SERS tags are simple colloidal nanoparticles (e.g., AuNP, AgNP, and Au–Ag alloys) or Au- or Ag-coated composites microspheres (e.g., SiO2@Au and SiO2@Ag) that usually face the problems of poor stability in complex samples, insufficient SERS activity, and uncontrollable hotspots (Jeon et al., Sens Actuators B Chem 2020, 321, 128521; Shen et al., J Hazard Mater 2022, 437, 129347; Zhang et al., Biosens Bioelectron 2018, 106, 204-211). These problems all weaken the performance (accuracy, reproducibility, and stability) of SERS-ICA. Notably, the current reported SiO2@Au and SiO2@Ag tags are fabricated by coating SiO2 cores with Au or Ag shells continuously through chemical reduction, which involves a complex manufacturing process and presents poor reproducibility (Pang et al., Chemical Engineering Journal 429 (2022) 132109; Liu et al., Sensors and Actuators B 258 (2018) 365–372; Yang et al., J. Mater. Chem. C, 2020,8, 2142-2154). The reasonable customization of high-performance SERS tags with high stability and dispersibility, multiple and controllable hotspots, and simple preparation processes for ICA detection remains challenging.

Second, real samples (e.g., food, clinical, and environmental samples) should be diluted for chromatographic detection to avoid the matrix effects exerted by complex compositions (e.g., proteins, lipids, acidic components, and debris). This requirement results in the decreased sensitivity and accuracy of SERS-ICA and even causes the missed detection of the target.

In this work, we reported a simple but high-performance SERS tag for SERS-ICA, which paved the way for the wide application of SERS-ICA methods in the POCT field. The SiO2@20Au tag was simply fabricated by assembling of one layer of dense 20 nm AuNPs onto the surface of 180 nm SiO2 via PEI-mediated electrostatic interaction, which can provide strong SERS activity, excellent colloidal and chemical stability, good dispersity, and multiple surface sites for bacteria binding. The main performance (stability, SERS activity, reproducibility) of SiO2@20Au tag is superior than those of commonly used SERS tags (For example, Cheng et al., ACS Nano 2017, 11, 4926-4933; Wang et al., Anal. Chem. 2017, 89, 1163-1169; Zhang et al. Biosensors and Bioelectronics 106 (2018) 204–211; Blanco-Covián et al., Nanoscale. 2017, 9, 2051-2058). The established SiO2@20Au-SERS-ICA can achieve direct detection of S. pneumoniae in 20 min with a detection limit of 46 cells/mL. Moreover, the proposed SERS-ICA maintained its high accuracy, specificity, and stability in real biological samples. This behavior confirmed that the SiO2@20Au-SERS-ICA has great potential for the rapid and accurate monitoring of S. pneumoniae in respiratory tract samples.

The reviewer mentioned paper 2022 August: Magnetic Nanotag-Based Colorimetric/SERS Dual-Readout Immunochromatography for Ultrasensitive Detection of Clenbuterol Hydrochloride and Ractopamine in Food Samples (Same NPs and same ICA for detection of antibiotics) is our work to develop a Magnetic Nanotag-based ICA method for food additives detection.

The reviewer mentioned papers 2022 August: Nanogapped Fe3O4@Au Surface-Enhanced Raman Scattering Tags for the Multiplex Detection of Bacteria on an Immunochromatographic Strip (Different NPs for bacteria detection)” and “2022 September: Ultrasensitive multichannel immunochromatographic assay for rapid detection of foodborne bacteria based on two-dimensional film-like SERS labels (Similar technology for Bacteria detection)” are our works to develop a Magnetic Nanotag-based ICA method and membrane-like SERS tags-based ICA method for foodborne bacteria.

Therefore, the principle of the current work is completely different from these previous articles. We believe the proposed SiO2@20Au-ICA method is innovative and valuable for respiratory pathogens detection and has strong potential for POCT use.

  1. Authors must describe organic layer in deep. FTIR, TGA and DLS to check that antibodies are properly linked and yield....

Answer: Thank you very much for your careful review and kind advice. This is really a good suggestion and it helps a lot to make our manuscript much better. Due to the amount or weight of the antibody immobilized on the surface of nanoparticles is limited, the TGA analysis may not be suitable for determining the amount of antibody on the SiO2@20Au NPs. We have employed the FTIR and DLS to verify the successful modification of antibody onto SiO2@20Au surface. The related results have been added in the revised manuscript (page 12 and supporting information Fig. S4.

Fig. R2 (A) DLS distributions of SiO2@20Au (black) and antibody-modified SiO2@20Au tags (red). (B) FTIR spectra of SiO2@20Au (blue line), anti-S. pneumoniae antibody (green line), and antibody-modified SiO2@20Au tags (red line). The characteristic absorption peaks corresponding to protein amide bands I (1641 cm1) and II (1530 cm1) appearing in immuno-SiO2@20Au reveals the success of coupling.

  1. Authors describe "obvious absorption peak at 531 nm". Since SERS is optimum if SPR of NPs is close to laser wavelength, why authors choose these gold NPs?

Answer: Thank you very much for your careful review and professional advice. As the review indicated, the SPR (~531 nm) of SiO2@20Au NPs is close to the excitation wavelength of 532 nm. As the reviewer indicated, if the SPR of SERS tags can match the provided excitation wavelength, the SERS activity of the nanotags further improved the resonance enhancement effect, resulting in higher detection precision and sensitivity (Bodelón et al. Nature Materials 2016,15, 1203-1211”, “Bai et al. ACS Appl. Mater. Interfaces 2014, 6, 3331−3340”, “Zhang et al. Food Chemistry 2015,169, 80–84”, “Guo et al. Nanoscale, 2015, 7, 2862). However, in fact, 532 nm incident laser was not commonly used in SERS-based ICA methods because the fluorescence background of strips is quite high under the excitation of 532 laser, which will destroy the quantification of Raman signal (Zhang et al. Biosensors and Bioelectronics 106 (2018) 204–211; Zhang et al., Sensors & Actuators: B. Chemical 277 (2018) 502–509; Shen et al., J. Mater. Chem. C, 2020,8, 12854-12864).

On the other hand, the 785-nm incident laser is the most commonly used for the detection of biological samples (such as bacteria, tumor cells, virus, and biomarkers) due to the sample damage issue reported in published literature (Guven et al., Analyst 2011, 136, 740-748; Shi et al., J. Biomed. Opt. 2014, 19, 056014; Yang et al., Biosens. Bioelectron. 2015, 68, 350-357). As for bioanalysis applications, the 785 nm excitation wavelength is suitable to reduce the damage to biological samples and the autofluorescence background. Many SERS-based lateral flow assay chose 785 nm excitation wavelength to reduce the damage to biological samples and the autofluorescence background, such as “Anal. Chem. 2017, 89, 10104-10110”; “RSC Adv., 2016, 6, 112079”; “Biosensors and Bioelectronics 106 (2018) 204–211”; “Analyst, 2018, 143, 2115–2121; “ACS Appl. Mater. Interfaces 2019, 11, 19495-19505”; “Journal of Hazardous Materials 437 (2022) 129347”. In addition, the excitation depth of 532 nm is less than 785 nm, which means more SERS tags in the nitrocellulose membrane will be excited with 785 nm laser and this contributes to the high signal to noise ratio (SNR) needed for ultrasensitive assays.

In this work, SiO2@20Au-based SERS-ICA strip was proposed for S. pneumoniae detection, thus we also used the 785-nm incident laser to reduce the damage to biological samples and the autofluorescence background. Moreover, the SiO2@20Au SERS tags performed well with 785 nm laser as source in the SERS-ICA strip. Therefore, 785-nm incident laser was chosen for the SERS detection of bacteria in this study.

  1. Did authors take into account that y axis should not have units? If authors consider that units must be showed, Why some spectra start in 10000?

Answer: Thank you very much for your careful review and kind advice. The unit of y axis of the SERS spectra is the relative Raman intensity (a.u.). In this work, the SERS spectra were collected under 785 nm excitation (5 mW, 5 s), baseline subtracted and shifted vertically for visualization. This operation is widely used in SERS label-based immunoassays (For example, Shen et al., Journal of Hazardous Materials 437 (2022) 129347; Wang et al., ACS Appl. Mater. Interfaces. 2019, 11, 19495; Zheng et al., Chemical Engineering Journal 448 (2022) 137760).

According to the reviewer indicated, we have added one sentence to explain this issue (Page 10) to make our manuscript much clearer to the readers.

  1. In order to know the amount of gold which is linked to SiO2, authors must perform an ICP analysis.

Answer: Thank you very much for your careful review and kind advice. In this work, the SiO2@20Au tag was simply fabricated by assembling of one layer of dense 20 nm AuNPs onto the surface of 180 nm SiO2 via PEI-mediated electrostatic interaction. The TEM image (Fig. 1E), the energy-dispersive X-ray spectroscopy (EDS) elemental line scan (Fig. 1F) and EDS elemental mapping (Fig. 1G) results have revealed that the outer layer of dense AuNPs were uniformly distributed on the surface of SiO2 core, which clearly demonstrated the structural components of the proposed SERS tag. 

The amount of AuNPs loaded onto SiO2 can be determined by EDS spectroscopy. As revealed in Figure R3, EDS spectrum indicates the presence of obvious Si, O, and Au signals in the SiO2@20Au nanostructure with the corresponding elemental composition (atomic fraction) of 31.04%, 55.80%, and 13.17%, respectively, which is in accordance with the element mapping results in Fig. 1G. The related content has been added in the revised manuscript (Page 10) and supporting information (Figure S2).

Fig. R3 EDS data from a single SiO2@20Au tag. The Cu signal is from the Cu grids of the TEM sample. The inset shows the element compositions in SiO2@20Au structure.

  1. Figure 3c. Did authors used Cryo-TEM to see bacteria?? Metal stains?

Answer: Thank you very much for your careful review and kind advice. We did not use the Cryo-TEM and metal stains technique. The TEM images of bacteria were obtained using a Hitachi H-7650 microscope operating at 80 kV. Many previous works have demonstrated the bacterial cell can be clearly observed with common TEM (For example, ACS Appl Mater Interfaces. 2015, 7(37), 20919-29; ACS Appl Mater Interfaces. 2016, 8(31), 19958-67; J Mater Chem B. 2018, 6(22), 3751-3761; Food Chemistry 363 (2021) 130400; Biosensors 2022, 12, 942). We appreciate the referee’s suggestion very much and have added a more detailed explanation on this issue in the revised supporting information (S1.1).

  1. Figure 4c. Could authors explain why SERS spectrum of 0 cells/ml have the main peaks of bacteria, but with less intensity? Moreover, Figure 4c has 9 spectra, while Figure 4d only have into account 7 intensities. Please, explain it.

Answer: Thank you very much for your careful review and kind advice. Technically speaking, the SERS-ICA strip for target detection is an indirect detection, and the signal is reported by the DTNB-modified SiO2@20Au SERS tags. When the tested samples contained the target bacteria, the SiO2@20Au tags will bind to the bacterial surface and the formed SiO2@20Au-bacteria complexes were specifically identified and captured by the anti-S. pneumoniae antibody on the test line. Then, the SERS signal of the test line was measured for the quantitative analysis of-S. pneumoniae. As the review indicated, a rather weak SERS signal (~1331 cm−1) on the test line can be still detected when the S. pneumoniae concentration is 0 cells/mL. This weak SERS signal is derived from the residual SERS tags in the running solution and can be deducted as the background signal. Notably, this is a common phenomenon in the SERS-based ICA methods (For example, Jeon et al., Sensors & Actuators: B. Chemical 321 (2020) 128521; Wang et al., Sensors and Actuators B 270 (2018) 72–79; Eryılmaz et al., Analyst. 2019, 144, 3573-3580; Xiao et al., Biosensors and Bioelectronics 168 (2020) 112524). Based on the calibration curve in Figure 4d, the LOD of our method is calculated as the corresponding concentration to the value: yblank + 3 × SDblank, where yblank is the average value and SDblank is the standard deviation of the background SERS peak intensity at 1331 cm−1. Thus, the LOD of our SERS-ICA strip was estimated to be 46 cells/mL. We have added more content to explain this issue in the revised manuscript (page 14) to make our manuscript much clearer to the readers.

In addition, the Figure 4c and Figure 4d in our work showed the SERS spectra of tested strips that corresponding to different concentrations of S. pneumoniae and the corresponding calibration curve, respectively. We found a hook effect at test line when the S. pneumoniae concentration reach 106 cells/mL. From the measured SERS signals in Figure 4c, the SERS signal of test line for S. pneumoniae exhibited a wide dynamic range over four orders of magnitude (106-50 cells/mL). Thus, only 7 SERS intensities (exclude the signals of 106 and 0 cells/mL) were used to construct the calibration curve. Notably, the most of the published SERS-ICA methods adopt the same way to plot the calibration curve (For example, Journal of Hazardous Materials 437 (2022) 129347; ACS Appl. Mater. Interfaces. 2019, 11, 19495; Chemical Engineering Journal 448 (2022) 137760; J. Mater. Chem. C, 2020,8, 2142-2154; Sensors and Actuators B 270 (2018) 72–79).

  1. Most important issue. Section 3.5 describes real samples. Sputum with the addition in the laboratory of bacteria are not real sample. Authors must evaluate real patient samples in order to improve the novelty of the work. Moreover, authors must show not only a table in the ESI, but all SERS spectra in a Figure in the main text.

Answer: Thank you very much for your careful review and kind advice. In China, detecting real biological samples from patients need the approval from the Ethics Committee of Hospital and the signed informed consents from all the patients. It is difficult for us to obtain the patients’ biological samples and ethic approvals in the short term. We appreciate your understanding in this matter.

Actually, many reports have shown that the accuracy of the newly-developed biosensor in practical applications can be evaluated by spiking the targets (e.g., bacteria, virus, toxins) in the real samples (e.g., clinical samples, food samples or environment samples) (For example, Small. 2021, 17(25), e2100862; Anal. Chem. 2021, 93, 8362-8369; Food Chemistry 289 (2019) 708–713; Food Chemistry 363 (2021) 130400; Anal. Chem. 2022, 94, 10865-10873; Nat Commun. 2020, 11(1):267; Sensors and Actuators: B. Chemical 325 (2020) 128780). Thus, in this work, we also spiked the concentration-determined S. pneumoniae into the sputum sample from healthy volunteers to simulate real respiratory tract samples. The detection results (SERS spectra) for each sample were averaged to generate a reproducible signal and substituted into the established calibration curves to calculate the recoveries of S. pneumoniae in the real sputum samples.

According to the reviewer indicated, we have added the detection results (averaged SERS spectra) in the Supporting information (Fig. S6) to make our manuscript more rigorous.

We appreciate for Editor/Reviewers’ warm work earnestly, and hope that the correction will meet with approval. Once again, thank you very much for your comments and suggestions.

Round 2

Reviewer 3 Report

1. A brief SERS description would be necessary.  Recomends.

DOI:10.1155/2018/4127108

2. Concerning FTIR, Could authors explain why antibody peaks could not be assigned: 2800, 1600, 1100,1300 and 750? Why are only minor differences between the 2 nanocomposites?

3. Answer from authors: "Thank you very much for your careful review and professional advice. As the review indicated, the SPR (~531 nm) of SiO2@20Au NPs is close to the excitation wavelength of 532 nm." Nevertheless, in the answer, scheme 1 and supporting methodology, a 785 nm laser is described. Please confirm and discuss this fact. Authors must describe and discuss in the text as well. It would be nice if authors perform the same esxperiment iwth a 532 nm laser. 

4. Authors must change title of section (3.5). May be something like "SERS analysis on modified human samples" is more realistic. 

Author Response

Dear Editor and Reviewers,

Thank you for your letter and for the reviewer’s comments concerning our manuscript entitled “Electrostatic adsorption of dense AuNPs onto silica core as high-performance SERS tag for sensitive immunochromatographic detection of Streptococcus pneumoniae”. Those comments are valuable and very helpful for revising and improving our manuscript. We have studied the comments carefully and have made corrections which we hope to meet the reviewer’s request. The revised portions are marked in red in the paper. The main corrections in the paper and the response to questions brought up by the academic editor and reviewers are listed as below:

To referee 3

Questions:

  1. A brief SERS description would be necessary. Recomends.

DOI:10.1155/2018/4127108

Answer: We think the reviewer’s suggestion is really a good one and it helps a lot to make our manuscript more rigorous. A brief description about SERS has been added in the introduction section of the revised manuscript and the recommended reference has been cited.

  1. Concerning FTIR, Could authors explain why antibody peaks could not be assigned: 2800, 1600, 1100,1300 and 750? Why are only minor differences between the 2 nanocomposites?

Answer: Thank you very much for your careful review and kind advice. In this work, antibody molecules were modified onto the surface of SiO2@20Au-DTNB to bind target bacteria. Notably, considering the big size of SiO2@20Au NP (~220 nm), the amount of antibody conjugated onto SiO2@20Au tags is relatively low. Thus, only the main characteristic absorption peaks corresponding to protein amide bands I (1641 cm1) and II (1530 cm1) can be observed onto the antibody-modified NPs (Fig. S4B). Actually, this FTIR result of antibody-modified NPs is common in the published literatures. For example, Fig. 1 in “Qie et al., Anal Chem. 2019, 91(15), 9530-9537. ( Please see Fig. R1)”; Fig. S4 in “Zheng et al., Chemical Engineering Journal 448 (2022) 137760. (Please see Fig. R2)”. Our FTIR result is consistent with these literatures.

Fig. R1 The Figure is from “Fig. 1 in Anal Chem. 2019, 91(15), 9530-9537”.

Fig. R2 The Figure is from “Fig. S4 in Chemical Engineering Journal 448 (2022) 137760”.

  1. Answer from authors: "Thank you very much for your careful review and professional advice. As the review indicated, the SPR (~531 nm) of SiO2@20Au NPs is close to the excitation wavelength of 532 nm." Nevertheless, in the answer, scheme 1 and supporting methodology, a 785 nm laser is described. Please confirm and discuss this fact. Authors must describe and discuss in the text as well. It would be nice if authors perform the same esxperiment with a 532 nm laser.

Answer: Thank you very much for your careful review and kind advice. Several previous works have clearly demonstrated that 532 nm laser is not suitable for SERS-ICA strips due to the high fluorescence background of NC membrane under the excitation of 532 nm, which will destroy the quantification of Raman signal (For example, Zhang et al., Biosensors and Bioelectronics 106 (2018) 204–211; Wang et al., Analytica Chimica Acta 1128 (2020) 184-192; Wang et al., ACS Appl. Mater. Interfaces 2019, 11, 19495-19505). We have discussed this issue in the revised manuscript (page 12) to make our manuscript much clearer to the readers.

  1. Authors must change title of section (3.5). May be something like "SERS analysis on modified human samples" is more realistic.

Answer: Thank you very much for your careful review and kind advice. The title of section (3.5) has been revised as you suggested.

We appreciate for Editor/Reviewers’ warm work earnestly, and hope that the correction will meet with approval. Once again, thank you very much for your comments and suggestions.
